# Evolutionary Divergence of Phosphorylation to Regulate Interactive Protein Networks in Lower and Higher Species

**DOI:** 10.3390/ijms232214429

**Published:** 2022-11-20

**Authors:** Claude Pasquier, Alain Robichon

**Affiliations:** 1I3S, Université Côte d’Azur, Campus SophiaTech, CNRS, 06903 Nice, France; 2INRAE, ISA, Université Côte d’Azur, Campus SophiaTech, CNRS, 06903 Nice, France

**Keywords:** phosphorylation, protein networks, kinases, evolution of phospho-proteins, kinetics of assemblages, P-sites, kinase motifs

## Abstract

The phosphorylation of proteins affects their functions in extensively documented circumstances. However, the role of phosphorylation in many interactive networks of proteins remains very elusive due to the experimental limits of exploring the transient interaction in a large complex of assembled proteins induced by stimulation. Previous studies have suggested that phosphorylation is a recent evolutionary process that differently regulates ortholog proteins in numerous lineages of living organisms to create new functions. Despite the fact that numerous phospho-proteins have been compared between species, little is known about the organization of the full phospho-proteome, the role of phosphorylation to orchestrate large interactive networks of proteins, and the intertwined phospho-landscape in these networks. In this report, we aimed to investigate the acquired role of phosphate addition in the phenomenon of protein networking in different orders of living organisms. Our data highlighted the acquired status of phosphorylation in organizing large, connected assemblages in *Homo sapiens*. The protein networking guided by phosphorylation turned out to be prominent in humans, chaotic in yeast, and weak in flies. Furthermore, the molecular functions of GO annotation enrichment regulated by phosphorylation were found to be drastically different between flies, yeast, and humans, suggesting an evolutionary drift specific to each species.

## 1. Introduction

Kinases elicit the inhibition and activation of a large spectrum of biological events via the addition of phosphate groups to amino acid residues. The end of the process involves the action of specific phosphatases removing the phospho group by cleavage. This appears as an essential mode of intracellular transduction pathways triggered by outside signaling inputs [1,2,3,4]. The phosphorylation of amino acids on eukaryotic proteins targets mostly serine, threonine, and tyrosine [5]. However, more recently, histidine is also a candidate substrate for some kinases, which is likely underestimated because of the phospho-histidine escaped analytical procedures due to its intrinsic chemical instability [6]. Most membrane kinases in the metabolic pathways are tyrosine kinase receptors activated by extracellular peptides or protein association involved in cell/cell interaction. In contrast, cytosolic serine and threonine kinases are activated by the transduction of external activators into cells through the transient elevation of calcium, cAMP, IP3, or any other second messengers [1,2,3,4,5]. The phosphorylation cascade integrates multiple regulatory steps and extends the fine tuning of interconnected metabolic pathways. This cascade is a key phenomenon for propagating the signals that take place within cells that associate multiple partners to inhibit or activate numerous types of enzymatic activities or protein functions. Moreover, the hubs of scaffold proteins that spatially and temporally organize the functional protein assemblages in eukaryotic cells are intrinsically linked to phosphorylation events, thus defining a phospho–dependent interactome that regulates the interaction of multiple proteins and creates many specific functions [2,7,8]. Scaffold architectures are recruited depending on the stimuli through a signal transduction mechanism that involves membrane embedded receptors and adaptors acting together through docking sites. These protein assemblages are a dynamic combinatorial state that constitutes a transient signalome [7,8,9,10]. These transient complexes can be considered as microstates that will be dismantled after the activation process stops. Abundant literature documents biphasic activity and allosteric activation as a consequence of a cascade of phosphorylation events creating protein association within the scaffold [9,10,11,12]. The first explored example is the glycogen phosphorylase which shows a transition inactive/active form that is allosterically controlled by the presence of a phospho group on residue Ser14 [1]. All types of extracellular mediators, such as neurotransmitters, hormones, light, neuropeptides, and cytokines, activate specific patterns of phosphorylation inside cells [2,5,12]. Moreover, these phosphorylations are part of a more complex landscape with other covalent modifications, such as ADP ribosylation, acetylation, myristoylation, isoprenylation, tyrosine sulfation, and ubiquitinylation, that create an integrated crosstalk that is difficult to apprehend due to its complexity and poor amenability to experimental validation [4,13]. For instance, distal crosstalk that combines arginine methylation and phosphorylation has been documented where methyl arginine hampers a phosphorylation on a serine residue at a remote place, or the inverse in some other cases [14,15,16]. During this last decade, sophisticated analytical methods have been successfully performed using mass spectrometry, functional interaction traps, and affinity chromatography in order to determine the high-throughput phospho-proteome within many species, despite the difficulty of trapping some phospho sites due to their rapid turnover, low levels or rare proteins, and the limits of technical performance [3,17,18]. The authors assumed accordantly that 86% of all phosphorylated residues are serine, with 12% being threonine and only 1.8% being tyrosine [5,17,18].

On the other hand, a large-scale analysis of the constraint of ideal linear sequence motif(s) presented by the substrates of each known kinase using a systemic approach to individually modify the order of amino acids was performed under the “Phosphorylome Project” using yeast proteome microarrays [19,20]. The high-throughput of phospho-tyrosine detection in different biological extracts has revealed a large proportion of noncanonical P-sites, which can be explained by the steric modification of proteins in the assemblage context [21,22]. All these data are in accordance with the dynamic structure of the flexible substrates to adapt to efficient binding to kinase catalytic sites [23]. The complexity of the phosphorylation phenomenon is drastically magnified by the fact that many linear motifs presented by different proteins in a given configuration are often potential substrates for several kinases. Since a few decades ago, authors have noticed the link between phosphorylation of the substrate motif(s) presenting the best affinity for kinases and the induced biological activities [7]. The very distant kinases in their mode of biological functions, such as PKA, PKC, and CamK, turned out to have very close structural features, with overlapping phosphorylation sites [7]. On the other side, many ideal linear motifs are buried and/or embedded with little accessibility, and consequently, escaping phosphorylation occurs in all the cases of biological stimulation. The 3D conformation of proteins trapped and embedded in highly organized and transient complexes reveals non-canonical sequences that are unexpectedly exposed high-affinity sites to kinases. These scaffolds are dynamic and transient complexes, resulting in temporary assemblage upon extracellular ligand binding, which induces a cascade of signaling events. One example is the RAS G protein that activates MAPKKK (RAF kinase), which activates MAPKK (MEK1 kinase), which then activates MAPK (ERK kinase), with the overall process depending on the scaffolding matrix that maintains the binding of the partners together [24,25,26]. Consequently, the major phosphorylation events to which a given protein is confronted depend on tethering by docking sites, localization in cell compartments, and insulating partners, which limits or increases the accessibility of substrate and kinases and eliminates unwelcome encounters. All these elements are arguments that can be evoked to take into account these experimentally observed discrepancies.

During the last decade, powerful technologies have drastically advanced and allowed the realization of public databases in which the exact and exhaustive phospho sites found by the experimental procedures for multiple species models are quantified. These curated lists, which are perfectly annotated and searchable, represent a powerful tool for the comparison of phospho-proteome between species regarding the homologous conserved proteins through different taxa from lower to higher orders [27,28,29,30,31,32,33,34,35]. The curated data are accessible on multiple websites, such as PhosphoSitePlus [31,32,33], Phospho.ELM [27,28], phosphoPep [19], PhosphoNET [36], phosphoGRID [37], and dbPAF [18] websites. In the past, these database resources have reported phosphorylation sites in proteins from *H. sapiens*, *M. musculus*, *R. norvegicus*, *D. melanogaster*, *C. elegans*, *S. pombe*, and *S. cerevisiae*, and these numbers have gradually increased due to their complementation with more recent experimental datasets. The detailed annotations of proteins are accessible for each entry, and the duplication or the multiple redundancy between these separate databases coming from divergent experimental procedures constitutes a powerful method for cross examination. The CUCKOO workgroup at Huazhong University of Science and Technology has been working for some years on the development of an integrative database of protein phosphorylation in animals, plants, and fungi [18,38]. In 2021, they released a database called EPSD (http://epsd.biocuckoo.cn, accessed on 8 March 2022—[39]) that combines the information stored in dbPPT [38], dbPAF [18], and 13 additional databases, including PhosphoSitePlus [31,32,33], Phospho.ELM [27,28], UniProt (The UniProt Consortium, 2019) [35], PhosphoPep [19], PhosphoGRID [37], dbPTM [23], FPD [40], HPRD [34], MPPD [41], P3DB [42], PHOSIDA [30], PhosPhAt [43], and SysPTM [44]. Altogether, EPSD contains 1,616,804 experimentally identified phosphorylation sites in 209,326 phosphoproteins from 68 eukaryotic species. 

Extremely abundant literature has therefore documented the biology of phospho-proteome. Numerous individual orthologous phospho-proteins have been compared in different contexts and in different species. However, little is known about how phosphorylation operates with large scale genomes and the full spectrum of proteomes. Very rare studies have explored the overall interacting protein landscapes induced by thousands of P-sites together that coincidentally act with each other. In a reverse effort compared to the general trend, which consists of analyzing proteins one by one with their few partners, we investigated the idea that global phosphorylation analysis might unmask unsuspected biological principles. Therefore, we aimed to highlight whether massive and integrated P-sites coincidentally contribute to the creation of matrices of proteins without which their function would not be fulfilled. 

In this report, we used bioinformatics tools to investigate, using a wide genome scale, the full phosphoproteome of a few generic biological models with respect to major integrated functions. First, we explored whether the degree of phosphorylation of hortologs is conserved across phylogenetically distant species. Second, we determined the comparative rate of phospho-sites in hortolog proteins for each of these species. Then, we investigated the correlation between the degree of phosphorylation of proteins and the size of interacting networks in which they were a component, which we validated using a statistically significant *p*-value. The protein networks were described by the number of nodes, edges, and the average number of neighbors, according to the standard tools of bio-informatic analysis. The GO annotation of molecular functions was equally scrutinized based on the degree of phosphorylation of proteins. Finally, the integrated substrate maps specific to six different kinases families were graphically superimposed to highlight the significant overlapping of P-sites within protein networks. In addition, an extended analysis of the time course of phospho-proteome induced by EGF factor was performed as an example to show the time-dependent, specific, and changing networks. The multi components and systemic analysis reported in this work show substantial divergence between higher and lower organisms regarding the role of phosphate addition to orchestrate larger protein complexes and phosphorylation dependent GO annotation specificities. All these elements argue in favor of an evolutionary achievement in higher organisms that has selected phosphorylation for which more phospho-sites in proteins guide larger sizes of interaction networks. These multiple and complementary computational analyses highlight the evidence supporting a meta-organization of proteins in different modalities of matrix, orchestrated by thousands of P-sites coincidentally acting with each other. 

## 2. Results

Large scale proteome was scrutinized in different model species representing major orders and taxa to determine a survey regarding their respective phosphorylation. The distribution of the number of phosphorylation sites per 1000 amino acids was carried out, and the results were compared between species. We aimed to investigate whether the phosphorylation status was correlated with the size of proteins and whether differences occurred between lower and higher species. Proteome was also divided in two panels: homologous protein found in other species versus non-homologous proteins with any resemblance elsewhere to establish whether phosphorylation targeted mainly one group or the other. Results are summarized in Table 1. Briefly, the main information of the computational searching on the large-scale proteome indicated few unambiguously trends, including the following: the *Homo sapiens*, *Mus musculus* proteome are massively phosphorylated (90% and 80%, respectively), *Drosophila* and *C. elegans* are weakly phosphorylated (38% and 28%, respectively), *A. thaliana* was moderately phosphorylated, and *Danio rerio* presents a uniquely low percentage (9%). Finally, the yeast *Saccharomyces cerevisiae* was found, surprisingly, at the level of humans and mice (87%). The analysis on the subcategories (homologous versus non-homologous proteins) showed no trends in phosphorylation status in each model species, which suggests that the addition of phosphate is not related to an ancient and primitive mechanism linked to the conservation of proteins through the evolutionary drift of living organisms. Few more elements highlighted by the computational analysis were striking. Homologous proteins found phosphorylated in *Homo sapiens* and *Mus musculus* showed a high percentage of the total protein (85% and 69%, respectively), whereas this percentage fell drastically in *Danio rerio* (6%), and significantly in *Drosophila*, *C. elegans*, and *Arabidopsis* (16%, 14%, and 19%, respectively). This result should be underestimated by the fact that in humans, the homologous protein component is massively represented (the non-homologous component represents less than 10% of the total). Moreover, the phosphorylated proteins had a higher molecular weight than the non-phosphorylated ones, which was constantly observed in all the species. The average length for the non-phosphorylated versus phosphorylated was 189/590 for *Homo sapiens*, 286/612 for *Mus musculus*, 535/847 for *Danio rerio*, 408/582 for Drosophila, 349/542 for Arabidopsis, and finally, 275/556 for *Saccharomyces cerevisiae*. This strongly suggests that phosphate addition occurs in larger sizes of proteins likely orchestrating the functionality of subdomains. The distribution of phosphate groups added to the individual protein in the seven species in proposed in Figure 1.

For this purpose, two parameters were graphically plotted: the number of phosphorylated sites per 1000 amino acids protein units and the frequency of such events in the whole proteome. Figure 1 shows the log–log plot of the relationship between these two entities. Interestingly, the observed parallelism of the distribution of phosphorylation sites did not correlate with the distance between species in a phylogenetic tree. The three most phosphorylated species already found in Table 1, namely *H. sapiens*, *M. musculus* and *S. cerevisiae*, showed a very similar slope. The two least phosphorylated species, *D. rerio* and *C. elegans*, presented an equally similar slope. Finally, *A. thaliana* and *D. melanogaster*, the two medium phosphorylated species, showed the same profile. What is remarkable is that the similarity of the slopes on the log–log plot between the seven analyzed species suggests that the different orders are highly uniform and homogenous.

### 2.1. Conservation of Phosphorylation in Homolog Proteins of Phylogenetically Distant Species

The degree of conservation of the phospho-sites on the heterologous proteins was investigated between the seven species representing different orders and taxa by computational search and quantification based on a full scale proteome. For each pair of analyzed species, we calculated a Spearman correlation between the phosphorylation rate of homologous proteins (see Materials and Methods). Figure 2 summarizes the results presented in the form of a heatmap where correlations were considered strong for values > 0.6, moderate for values between 0.6 and 0.4 and weak for values between 0.4 and 0.2. All correlations were positive. At the top of the heatmap, the phylogenetic tree of the seven species is drawn. The importance of the correlation was roughly linked to the distance of the species on the phylogenetic tree. The only exception was *Danio rerio* which was, surprisingly, the species whose distribution of phosphorylation sites was the furthest from humans.

We expected to observe a correlation between the number of phosphorylation sites per protein and their lengths (or masses). We investigated whether this correlation between masses and the number of phosphorylation sites for each phosphorylated and orthologous protein was divergent within each species. Appendix A plots the regression line for each species as an index of phosphorylation density. The slopes of the regression lines were found in accordance with the distances in the phylogenetic tree, except for *Danio rerio*. To reduce the influence of outliers on the determination of the regression line, we zoomed the figure to select only proteins with a mass of less than 500,000 daltons and with less than 500 phosphorylation sites. Results, presented in Figure 3A, confirmed the marked differences between species.

### 2.2. Comparative Rate of Phospho-Sites in Homolog Versus Unique Proteins for Each of These Species

The computational analysis of full scale proteome was pursued to investigate the comparative analysis of phosphorylation in proteins that have a homolog in another species versus those without a homolog. Figure 3B and Appendix A show the data for *Homo sapiens*. This analysis confirms that proteins with homologs were much longer than those without. In terms of the distribution of phosphorylation sites, we noticed that proteins with homologs were slightly more phosphorylated than those without. The same observation applied to *Mus musculus* (see Figure 4A and Appendix A). The proportion of non-homologous protein is very low in these two species (humans and mice); therefore, a bias for rigorous interpretation is possible. For *Danio rerio*, the opposite was observed as proteins without homologs were longer than those with homologs, with similar distribution of phosphorylation sites (see Figure 4B and Appendix A). The same observation related to *Danio rerio* applied to *Drosophila melanogaster* (Figure 4C and Appendix A). For *Caenorhabditis elegans*, the proteins with homologs were shorter and also had fewer phosphorylation sites (see Figure 4D and Appendix A). For *Saccharomyces cerevisiae*, proteins without homologs were shorter than those with homologs and had significantly more phosphorylation sites (see Figure 4E and Appendix A). For *Arabidopsis thaliana*, a pattern where the proteins without homologs were longer than the others with the regression lines almost in superposition was observed (see Figure 4F and Appendix A). Altogether, these data argue in favor of an evolutionary drift inside each species that exhibits, to some extent, little coherence and a chaotic history.

### 2.3. Correlation between the Degree of Phosphorylation of Proteins and the Size of Interacting Networks

Comparative evolutionary analyses based on computational tools were performed to extract phosphorylation information to uncover the role of phosphate addition in mastering a larger network of interacting proteins. For *Homo sapiens*, we divided the 19,645 proteins in our dataset into deciles according to the number of phosphorylation sites. Thus, there were 2368 proteins with zero, one, or two phosphorylation sites in the first decile, 2130 proteins between three and five phosphorylation sites in the second decile, etc. (see Table 2).

The interactions between proteins in the same decile were calculated using the String database and a confidence score greater than 0.7. This determined the number of interactions between proteins (edges), the average number of interactions of each protein (avg. Neighbors), the number of connected components (nb. Connected), and the percentage of proteins with no interactions (% non-connected nodes). The graphical representation of the average number of interactions for each decile is given in Figure 5A.

We noticed that the average number of interactions was correlated with the number of phosphorylation sites (or the more phosphorylation sites there were on a protein, the more interactors it had). In humans, the massive networks measured by the number of edges, nodes, and averaged neighbors were found to be clearly dependent on and correlated with the increased number of phospho-sites per protein.

We chose to do the same type of analysis for two other species: *D. melanogaster* and *S. cerevisiae*. *D. melanogaster* contains a large proportion of proteins without a phosphorylation site (more than 60% of the total), which makes it impossible to divide into deciles. We chose to consider the non-phosphorylated proteins as a category and to split the rest into quartiles. The data for *D. melanogaster* are specified in Table 3. We noticed a relative correlation between the average number of interactions and the number of phosphorylation sites, but this trend does not seem highly significant compared to *Homo sapiens*. The graphical representation of the average number of interactions for each category of phospho site density is given in Figure 5B (the high number of interactions obtained for the non-phosphorylated proteins might be biased because there are seven times more proteins in this category). For *S. cerevisiae*, the phospho-proteins was divided into quintiles (see Table 4). In this organism, a relative correlation between phosphorylation sites and interactions was observed, although the quintile of the most phosphorylated proteins showed a relative decrease. As for humans, the number of edges and neighbors increased with phosphorylation and were inversely correlated with the number of connected components for these two species (see Table 4 and Figure 5C).

### 2.4. GO Annotations Analysis Based on the Degree of Phosphorylation of Proteins

The same categories defined by the degree of phosphorylation (including deciles for humans and quintiles for flies and yeast) were statistically assigned to a GO enrichment under the terms of a biological process. We found that in humans, non-phosphorylated proteins (those in the first two deciles) have a role in the immune response and perception. The most phosphorylated proteins (belonging to the last four deciles) have multiple regulatory roles (transcription, cell cycle, cellular response, splicing, and chromatin remodeling) and functions related to development and morphogenesis. All the significant annotations (which are associated with an adjusted *p*-value < 0.05) are listed in Appendix A. Overall, we saw that significant annotations were much more numerous for the deciles corresponding to the most phosphorylated proteins, including: 780 for decile 10, 369 for decile 9, 21 for decile 8, 15 for decile 7, and two for decile 6 were found. No significant enrichment was found for deciles 4 and 5, and there were three significant annotations for decile 3 and 2, and 15 for decile 1. Appendix A contains a list of selected annotations for humans. A visualization of the *p*-values associated with the selected annotations for each of the deciles is proposed as a heatmap in Figure 6.

In flies and yeast, paradoxically, there were more annotations on the list of no or low phosphorylated proteins than on the most phosphorylated ones. For yeast, only the first two quintiles (the least phosphorylated proteins) were significantly annotated with GO terms. Significant annotations for humans did not emerge for flies and yeast. However, there were shared annotations between these two latter organisms like the ones corresponding to mitochondrial translation. Remarkably, in flies and yeast, we found GO annotations mostly in the low or very low-level quantiles of phosphorylation, which appeared to be the opposite in humans. All the significant annotations for *D. melanogaster* and *S. cerevisiae* are listed in Appendix A respectively. The corresponding heatmaps are in Figure 7 and Figure 8.

### 2.5. Integrated Substrate Map for Six Different Kinases Families Highlights a Significant Overlapping

The site http://kinase.com/wiki/index.php/Kinase_classification (accessed on 18 March 2022) was used to perform a survey based on the full spectrum of proteomes to determine the extent to which the kinases had overlapping targets. Among these 14 groups of kinases, eight had identified targets that were listed in publicly available databases for different species. The number of phosphorylated proteins by this later group is listed in Appendix A. The whole dataset represents a total of 7073 proteins. From the list in Appendix A, we distinguished that six kinases had more than 3000 targets (AGC, CAMK, STE, CMGC, TK, and CK1) and three kinases had less than 1500 ones (TKL, Other, and Atypical). Plotting of the phosphorylated proteins by the six major kinase groups in a Euler diagram representation shows an intertwined complex network which mixes strict specificity and dense overlapped targeting. The representation is shown in Figure 9.

A large number of proteins were phosphorylated by all kinase groups, but the TK (Tyrosine Kinase) group differed from the others due to the fact that it phosphorylated restrictively and specifically most of its proteins targets that were insensitive to the action of the other kinase groups. In order to simplify this very dense graphical representation, we kept only the most prominent kinases, i.e., those that target more than 2% of the total proteins. By doing this, we obtained a drastic simplification. We noticed that the STE, CK1, and CAMK groups phosphorylated exactly the same group of proteins. The same is true for a group composed of Other, TKL, and Atypical. By regrouping these clusters, their overlapping, and their intersection, we propose a simplified graph that highlights the intertwined landscape of interconnected phosphorylation processes (see Figure 10). Figure 10 shows that some proteins were only phosphorylated by kinases belonging to the TK, CMGC, and AGC categories. Two hundred and twenty-four identified proteins (in the center of the diagram) were phosphorylated by all kinase groups. Interestingly, the targets of Other, TKL, and Atypical kinase groups were also targets for all the other kinases with any restrictive specificity. The two remaining groups of proteins were one group phosphorylated by all kinase groups except Other, TKL, and Atypical and the other group phosphorylated by STE, CK1, CAMK, CMGC, and AGC.

To further investigate the clustering of the Figure 10, a GO annotation for enrichment analysis was performed with the corresponding lists (see Appendix A). The functions of AGC-phosphorylated proteins are more diverse and concern biological regulation (FDR = 2.6 × 10^−4^), signaling (FDR = 1.2 × 10^−3^), response to stimulus (FDR = 1.2 × 10^−3^), or localization (FDR = 4 × 10^−3^). Highly significant annotations were found for proteins phosphorylated by all kinase groups, such as the negative and positive regulation of cellular process (FDR < 1 × 10^−46^), signal transduction (FDR = 8.75 × 10^−41^), response to organic substance (FDR = 1.40 × 10^−46^), or cellular component organization (FDR = 1.86 × 10^−37^). Proteins phosphorylated by all groups of kinases except Other, TKL, and Atypical had few significant annotations among; therefore, we can mention cytoskeleton organization (FDR = 2.3 × 10^−4^). Proteins phosphorylated by STE, CK1, CAM4, CMGC, and AGC also had few significant annotations, such as chromatin organization (FDR = 3.68 × 10^−3^) or RNA processing (FDR = 3.68 × 10^−3^).

### 2.6. Time Course of Phospho-Proteome and Time Dependent Specific Changes in Networks Induced by EGF Factor

Finally, we compared changing and non-changing subsets of the phospho-proteome using as example human Phospho-sites that were unaffected or affected by the stimulation of EGF in a culture cell system in a series of time course-controlled experiments. The aim of this study was to visualize the changes in the networks of correlated and interactive proteins upon a stimulus action in a time dependent manner. Data relative to the variation of phosphorylation following EGF exposure on Hela cells were measured using SILAC (stable isotope labelling by amino acids in cell culture) at six time-points: after 1 min, 5 min, 10 min, 15 min, 20 min, and 30 min, were retrieved from the qPhos database. These data were then processed to associate each protein with a count that corresponds to the variation in phospho-sites at each time point (see Materials and Methods). The summary of the number of phosphorylations per protein for each time point is specified in Appendix A. As a function of time after EGF exposure, the change in the number of phosphorylations per protein ranged from zero (no change) to 20 (the maximum measured for the Desmoyokin protein—AHNAK) after 10 min. Using the data in Appendix A, it is easy to list, for each time point, the proteins that were the most differentially phosphorylated. However, these lists highlight potentially unrelated proteins from which it is difficult to infer a meaning. In transcriptomics, following Rapaport et al.’s 2007 statement [45] that “a small but coherent difference in the expression of all the genes in a pathway should be more significant than a larger difference occurring in unrelated genes”, the focus has shifted to analyzing the activity of genes in the light of our knowledge of their molecular interactions, i.e., researchers seek to identify dense parts of an interaction network that contain genes that are significantly dysregulated. Those parts of an interaction network that are both dense and active are called “modules” or “active modules”. In the present study, we took a similar approach by identifying the most differentially phosphorylated protein modules on a protein–protein interaction network. These modules of interacting proteins differentially phosphorylated at each time point were identified using AMINE (Active Module Identification through Network Embedding—[46]) and listed in Appendix A. Depending on the time elapsed since the exposure to EGF, the number of modules varied from three at 1 min to a maximum of 18 at 15 min and decrease to two at 30 min. The structure of the modules was similar to a large one composed of several proteins ranging from 17 to 118 and other smaller ones. In this study, we focused on the first module. The enrichments of this module with annotations from Gene Ontology (Biological Process), KEGG, Reactome, and WikiPathway for the different time points are shown in Appendix A. Module 1 contains, for all the time points, the three proteins EGFR, SHC1, and MAPK1. Overall, there was a large overlap of proteins belonging to module 1 at all the time points. The evolution of the phosphorylation of the proteins included in the prominent module 1 representing protein interaction for each time point is shown in Figure 11. The proteins in red and those in brown were differentially phosphorylated compared to the control. The proteins in red represent those that were differentially phosphorylated at this time point compared to the previous time point (or newly phosphorylated proteins). The proteins in brown are those that were differentially phosphorylated but were already differentially phosphorylated at the previous time point. Finally, the proteins with a white background are those that were no longer differentially phosphorylated at this time point but were differentially phosphorylated at the previous time point (or back to normal). This analysis highlights the dynamic protein networking upon external stimulation. A dynamic and variable landscape of networks linked to stimulus-specific phospho-proteome can be considered as the rewiring of a signaling network.

## 3. Discussion

Phosphorylation constitutes a major post-translational modification ubiquitous in prokaryotic and eukaryotic cells, inducing transformation in protein steric configuration and activation and inhibition of various enzymatic activity and promoting and impeding protein–protein interactions. Consequently, phosphorylation orchestrates many biological processes, acting as a signal transduction triggered by extracellular factors. As expected, aberrant phosphorylation has been described in numerous reports as the cause of a large number of diseases, such as cancer and dysfunctional developmental programs across species [47]. Alzheimer disease in humans has been particularly scrutinized [48,49,50], along with other neuronal afflictions [51]. The immense effort to build a comprehensive inventory of the phospho-proteome across species in order to document the phosphorylation across all the spectrum of eukaryotic and prokaryotic living organisms has been undertaken in the past two decades [52,53,54,55,56]. The comparative catalog of this inventory amenable to the full genome scale of protein byproducts in key species could deepen our comprehension of this important biological mechanism that has been selected in evolution for five hundred millions years. Published advances have facilitated our evolutionary analysis, including searching highly conserved mechanisms that might have spread over living organisms and the singular derived events within each species [57]. 

Regarding the prokaryotes, or the most primitive organisms, only a few hundred proteins per organism have been reported to be phosphorylated, whereas for the eukaryotic species, the detection of thousands of phosphorylated proteins have been comparatively found. Bacteria have been intensively studied due to the simplicity of cells amenable to experimental procedures [58,59,60]. In this report, an evolutionary analysis was conducted to investigate whether a core of phosphoproteins is preferentially conserved in all species from lower to higher evolution hierarchy. We aimed to document whether protein orthologs across living organisms might share a hidden prominent mark like a signature that would be conserved in phosphorylated sites. However, when the phosphoproteome of yeast and *Caenorhabditis elegans* are compared, the results suggest that phosphorylation evolved separately after the divergence of higher eukaryotes from yeast, which is corroborated by the apparition of a large number of species-specific kinases after the split [61]. Interestingly, mitochondria have a bacterial origin in accordance with the symbiotic paradigm and are found in pro and eukaryotic cells. Their phosphorylation status shows unambiguous, limited conservation, which seem to indicate that phosphorylation is a more recent process evolving separately inside each taxa and orders. Moreover, the quantification of phospho-sites in mitochondria in diverse species shows a drastic elevation in mammals compared to that of yeast [61]. 

An analysis of phosphoproteome between non-heterolog and heterolog proteins in humans was informative—no clear-cut status can dissociate the two groups, although the non-ortholog proteins constitute a very small component versus the total. Within the eukaryotic reign, identified phosphoproteins are no more orthologs than non-orthologs depending on the species. 

Interestingly, in flies and worms, which have roughly the same phylogenetic distance to mammals, the same low level of phosphorylation associated with overlapping GO enrichment was observed. This pattern of GO annotation obtained in a comparative analysis between humans on one side and worms and flies on the other side is drastically different, suggestinglittle conservation of the phosphorylation and its biological implication across evolution. 

Together, comparative large scale phosphorylation datasets in yeast, flies, mice, and humans suggests striking differences in structural features. The intensity of phosphorylation is correlated with increases in the size of interacting networks in humans, whereas this trend is unclear or perhaps non-existent in lower species like flies. This discrepancy between the conservation of alignments in protein sequences and the evolutionary divergence in the phosphorylation process indicates that each species has manipulated the regulatory processes of phosphate addition in a very specific manner. In accordance with this finding, the size of the worm kinome is two times larger than of that of the fly kinome, with a high level of divergence between them as half of worm kinone are specific to this species [61,62,63]. The fact that 90% of human proteins are phosphorylated highlights the evolutionary success of primates in creating large, connected protein networks whose transient association and dissociation is obviously guided by phosphorylation [64]. Primates have invented robust phosphorylation processes to integrate multiple connected partners, which is not seen in other species, such as flies, zebra fish, and worms. However, in E. coli, for example, around 100 of its 4000 (3%) proteins are known to be phosphorylated, versus more than half of all proteins in eukaryotic cells [61]. Site-specific phosphorylation co-evolved with the adaptation of the species according to their evolutionary history. This suggests that regulation via phosphorylation occurred late in prokaryotic evolution from the split between yeast and higher eukaryotes. This might explain why eucharyotic species far remote in the phylogenetic tree share some common features regarding the kinone [61,65,66,67,68].

Based on one example of biological stimulus acting on a simple cultured cell, we present a comprehensive time course of protein networks triggered by time dependent changes of phospho-sites. After stimulation, the networks arranged by phosphate addition extend in a wave like manner and then retract to the basal level.

It is generally accepted that amino acid changes in protein sequences have evolved at a constant rate in history within each species. Phylogenetic analysis based on genome sequencing and RNA-sequencing determination have shown that the switch between acidic and phosphorylatable residues took place for some proteins in the evolutionary tree between eukaryotes and prokaryotes. Thus, the acidic residues (Asp or Glu) might have been replaced by serine or threonine—in such a case, the salt bridge can be restored via the addition of a phosphate group. In this scenario, the proteins authorizing chemical modification due to the phosphate addition are multi-phasic, allowing fine regulation [8]. In this report, we tried to address when an ordered apparition of kinases took place in the evolutionary timing of living organisms, and some major events of phosphorylation could explain the primate status [67,68]. 

## 4. Conclusions

In this report, we did not focus our study on known substrate motifs as signaling modules across species for major kinases. We assumed that in the evolutionary processes, some motifs can appear with modified kinases, diminishing the pertinence of identical motifs searching in full scale phosphoproteome. Instead, we preferred the strategy of determining the global phosphorylation in proteins and its incidence to guide molecular networks. 

Few, if any, comparative quantifications of overall phosphorylation and its evolutionary conservation across species have been undertaken. Using bioinformatics tools on genome wide scale eukaryotic phosphoproteomes, including humans, flies, worms, and yeast, we present the most comprehensive evolutionary landscape of eukaryotic phosphoproteomes thus far. We scrutinize the phosphoproteomes in these different models of generic models for which extensive full scale genome datasets are available. Most reports describe the phosphorylation in individual proteins along with their few regulatory partners and their comparison between species. These studies are usually also establish the degree of evolutionary conservation. Our approach documents a new paradigm: the role of the integrated P-sites landscape, including phosphorylation noise without known biological function, in the shape and assembly of proteins networks. Taking into account the massive P-sites without any known function (only 3% to 5% of P-sites have been found with a regulatory function), we explored the relationship between the full spectrum of phosphorylation and the meta organization of protein matrixes. A panel of computational tests addressed the role of full spectrum P-sites, including: (i) in relation to the dynamics of protein networking upon stimulation; (ii) in relation to the conserved networking between species; and (iii) in relation to the landscape of intertwined P-sites by different kinases into the networks. More importantly, our datasets highlight that P-sites numbers in proteins, including mostly “noisy” phosphorylation, increase drastically with the size of networks in which these proteins are component. These datasets might unmask a prominent role of the “P-site noise”, which is attributed to P-sites without a known function that constitute more than 95% of the total, regarding shaping proteins matrixes. 

We might conclude that phosphorylation orchestrates protein networks in humans. Amazingly, this observation appears to be more chaotic in lower species. The functional role of many phospho-sites remains unknown and poorly documented despite a considerable inventory of phosphorylated proteins with sequences information of modified residues across many species, ranging from lower to higher organisms. In the massive background and noise of phosphate addition, only a few highly conserved phosphorylation hotspot regions have been documented with functional importance [69]. If 90% of human proteins are found phosphorylated, only 5% have a reported regulatory function situated mostly in the catalytic domain or the protein–protein interaction domain [67,68,69].

Overall, all these imbricated studies highlight that the kinome and phospho landscapes evolve separately in each species, with a mixed of conserved processes between them and innovative new kinases and P-sites specific to each one. Our overall analysis at full proteome level demonstrates clearly that the paradigm of the great proportion of phosphate addition without any known function and considered metabolic noise should be re-evaluated. Phosphate addition, besides the known regulation of biochemical function regarding few individual proteins, likely intervenes with the orchestrating network of interacting proteins upon stimulation through what is unanimously viewed as its noise component. Moreover, stable or long-lasting P-sites without any attributed function might be a chemical key to assemble proteins in networks. 

## 5. Materials and Methods

### 5.1. Data Gathering

Phosphorylation data for seven model organisms (*Homo sapiens*, *Mus musculus*, *Danio rerio*, *Drosophila melanogaster*, *Caenorhabditis elegans*, *Saccharomyces cerevisiae*, and *Arabidopsis thaliana*) were downloaded from the EPSD database (http://epsd.biocuckoo.cn/, accessed on 8 March 2022). The reference lists of proteins for the same seven organisms was retrieved from the Uniprot database (https://www.uniprot.org/, accessed on 6 March 2022). Homology data between the proteins of these organisms were downloaded from the Homologene database (https://ftp.ncbi.nih.gov/pub/HomoloGene/current/homologene.data, accessed on 5 March 2022). An updated Excel table of the catalog of kinase proteins built by Manning et al. [70] was downloaded on March 18, 2022 from http://kinase.com/human/kinome/tables/Kincat_Hsap.08.02.xls, accessed on 8 March 2022. Data concerning the dynamics of phosphorylation were downloaded from the Qphos database (http://qphos.cancerbio.info, accessed on 20 May 2022—[71]).

### 5.2. Computing Correlation between the Distribution of the Number of Phosphorylation Sites Per Protein in Different Species

HomoloGene is a system that automatically detects homologs, including paralogs and orthologs, among the genes of 21 completely sequenced genomes. Homologene database attributes a HID (HomoloGene group ID) to proteins from different species that represents a grouping of homologous genes among species. This data has been combined with the number of phosphorylation sites per proteins given by EPSD. For each pair of analyzed species, we calculated a Spearman rank correlation, which gave a value in the range [−1, +1] where a value of +1 meant a perfect positive association between ranks, a value of 0 meant no association between ranks, and a value of −1 meant a perfect negative association between ranks. Data about the phylogenetic tree of the seven selected species were from Hedges (2002) [72].

### 5.3. Correlation between Phosphorylation and Protein Interactions

For humans, we divided the 19,645 proteins in our dataset into deciles according to the number of phosphorylation sites. There were 2368 proteins with zero, one, or two phosphorylation sites in the first decile, 2130 proteins with between three and five phosphorylation sites in the second decile, etc. (see Table 2). For *S. cerevisiae*, we divided the data into quintiles (Table 4). *D. melanogaster* contained a large proportion of proteins without a phosphorylation site (more than 60% of the total) which made it impossible to divide it into bins of equal size. For this organism, we considered the non-phosphorylated proteins as a category and sub-divided the rest into quartiles (Table 3). The interactions between proteins in the same bin were retrieved from the String database and used a confidence score greater than 0.7. Calculations of the number of interactions, the average number of interactions of each protein, the number of connected components, and the percentage of proteins having no interaction were done with Cytoscape [73].

### 5.4. Enrichment Analysis

Functional enrichments for Gene Ontology Biological Process terms, KEGG, WikiPathways, and Reactome Pathways were retrieved using Cytoscape [73] and the StringApp plugin [74]. Only terms associated with a False Discovery Rate (FDR) of less than 0.05 were retained.

### 5.5. Study of Phorphorylated Proteins According to the Kinase group

The site http://kinase.com/wiki/index.php/Kinase_classification (accessed on 18 March 2022) was used to perform the survey. Euler diagrams shown in Figure 8 and Figure 9 were generated using the online tool nVenn available at http://degradome.uniovi.es/cgi-bin/nVenn/nvenn.cgi (accessed on 20 April 2022) [75].

### 5.6. Dynamic Analysis

In the qPhos database, we selected data relative to the variation of phosphorylation following EGF exposure to Hela cells measured using SILAC (stable isotope labelling by amino acids in cell culture) at six time-points: after 1 min, 5 min, 10 min, 15 min, 20 min, and 30 min. The dataset contained the variation of phosphorylation at each time-point compared to the control for multiple phosphorylation sites. For the same protein, increases of phosphorylation were observed on some sites, while some other sites showed a decrease. In order to obtain a unique value reflecting its change in phosphorylation for each protein, we counted the number of sites where the variation in phosphorylation exceed a factor of two (a Log2 FoldChange EGF vs. control ≤ −1 or ≥ 1). Our count of the number of phosphorylation variation by protein for every time point is specified in Appendix A. Modules of interacting proteins differentially phosphorylated at each time point were identified with AMINE (Active Module Identification through Network Embedding—[46]). The method was adapted to use the number of phosphorylation variation per protein computed above instead of a *p*-value generated by a differential analysis pipeline. The String database [52] was used to retrieve protein interaction data with a combined evidence score greater than 0.7. Measures of protein activity were represented by phosphorylation change counts.

## Figures and Tables

**Figure 1 ijms-23-14429-f001:**
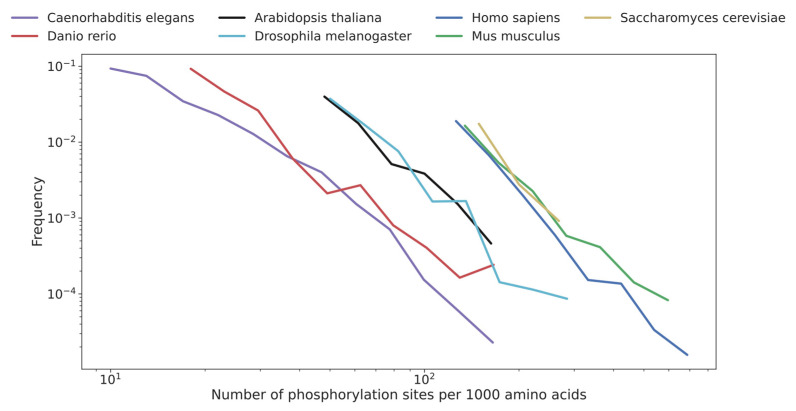
A log–log plot illustrating the relationship between the number of phosphorylation sites per 1000 amino acids and its frequency for seven species. The figure shows a clustering of species into three groups that share similar phosphorylation rates and have a very close distribution of phosphorylation sites.

**Figure 2 ijms-23-14429-f002:**
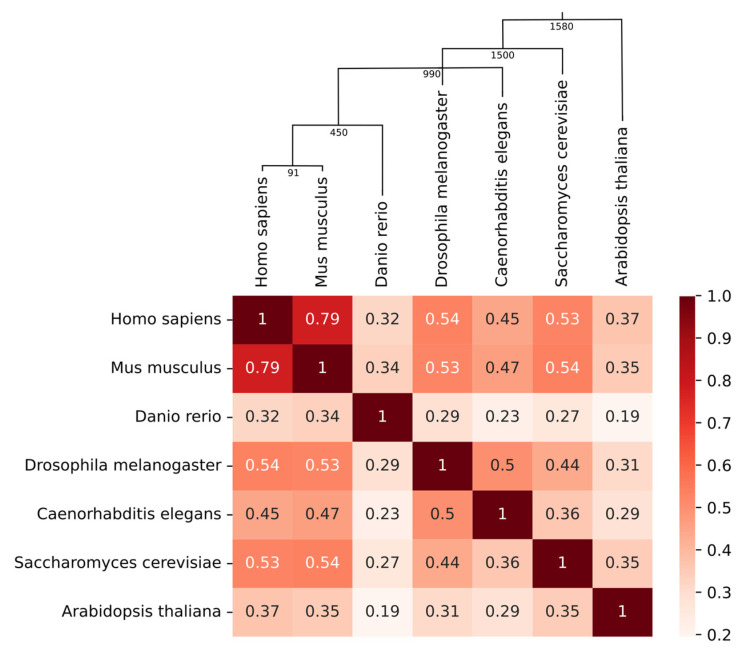
A heatmap illustrating the Spearman correlation between the phosphorylation rate of holologous proteins. At the top of the heatmap, the phylogenetic tree of the seven species is drawn. Times of divergence (million years ago) are indicated at the nodes in the tree.

**Figure 3 ijms-23-14429-f003:**
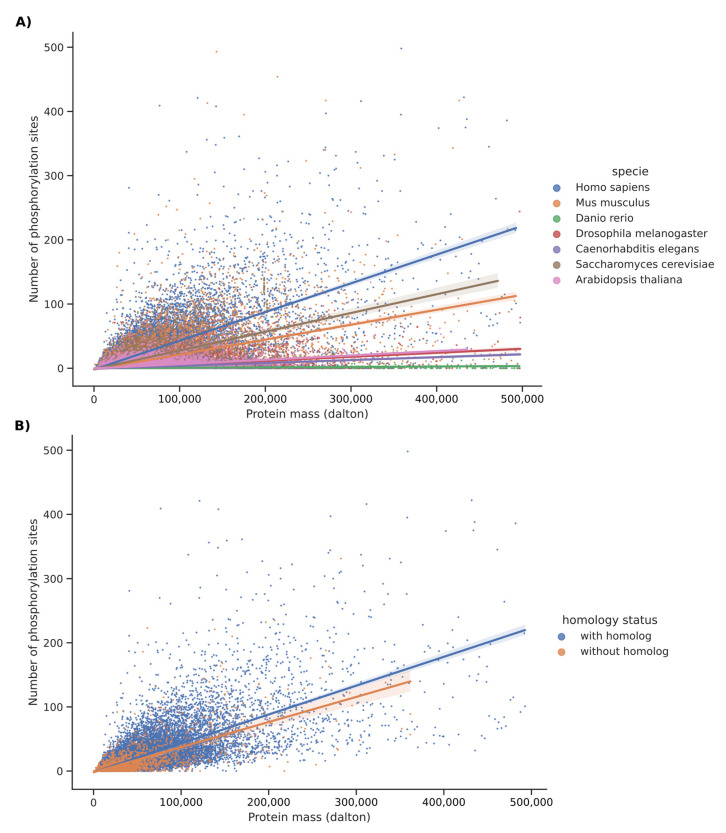
A plot of the relationship between masses and the number of phosphorylation sites. The study was carried out for each phosphorylated protein with a mass of less than 500,000 daltons and with less than 500 phosphorylation sites. (**A**) A plot regrouping the seven studied species. Regressions lines are drawn with a confidence interval of 0.95 (highlighted using translucent bands around the regression line). (**B**) A plot of phosphorylated proteins in *Homo sapiens*. Data in blue are for proteins that have at least one homolog in another species, while data in orange are for proteins without homolog. Regressions lines are drawn with a confidence interval of 0.95 (highlighted using translucent bands around the regression line).

**Figure 4 ijms-23-14429-f004:**
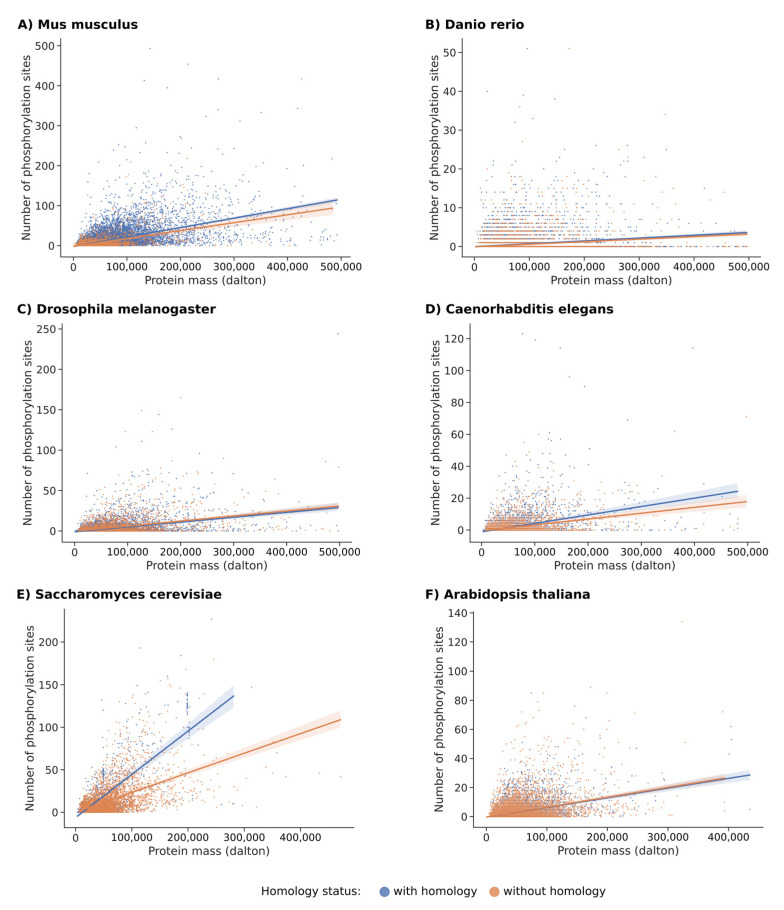
A comparative plot of the relationship between masses and the number of phosphorylation sites for six species of different order. Each phosphorylated protein with a mass of less than 500,000 daltons and with less than 500 phosphorylation sites was considered. Data in blue are for proteins that have at least one homolog in another species, while data in orange are for proteins without homolog. Regressions lines are drawn with a confidence interval of 0.95 (highlighted using translucent bands around the regression line). The species under consideration are (**A**) *Mus musculus*, (**B**) *Danio rerio*, (**C**) *Drosophila melanogaster*, (**D**) *Caenorhabditis elegans*, (**E**) *Saccharomyces cerevisiae*, and (**F**) *Arabidopsis thaliana*.

**Figure 5 ijms-23-14429-f005:**
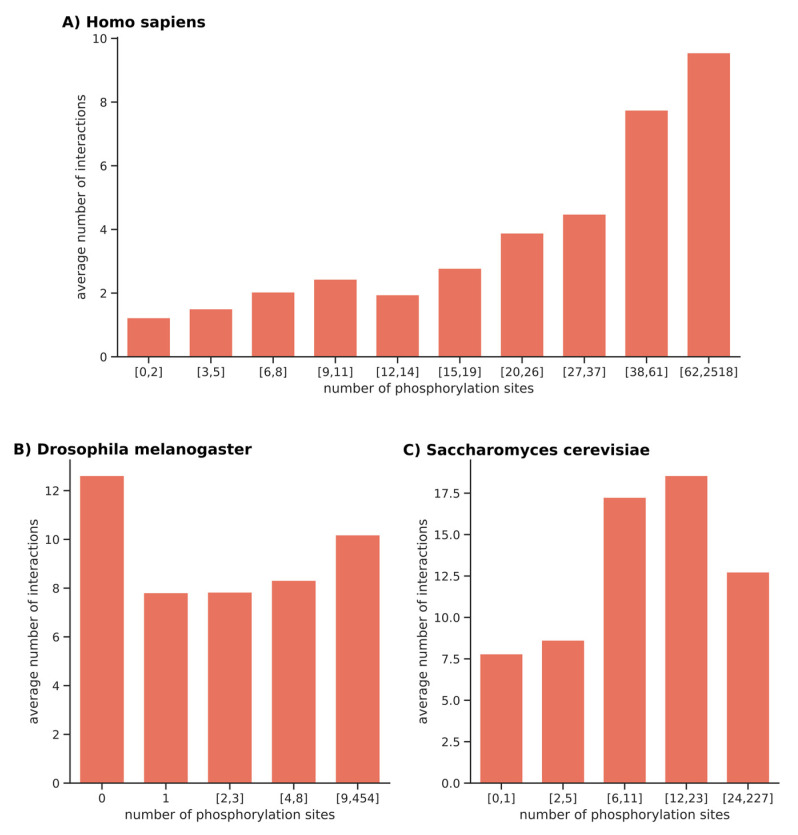
Average number of interactions per protein in function of its number of phosphorylation sites. Each bar represents a bin. The closed intervals, in terms of the number of phosphorylation sites defining a bin, are given on the x-axis. The average number of interactions for proteins belonging to each bin is given in the y-axis. (**A**) For *H. sapiens*, each bar represents a decile. (**B**) For *D. melanogaster*, the non-phosphorylated proteins constitute the first bin, and the remaining proteins are divided into quartiles. The height of the first bar, which represents the category of non-phosphorylated proteins, cannot be compared to the others because it contains seven times more proteins and the probability of obtaining interactions in this network is much higher. (**C**) For *S. cerevisiae*, each bar represents a quintile.

**Figure 6 ijms-23-14429-f006:**
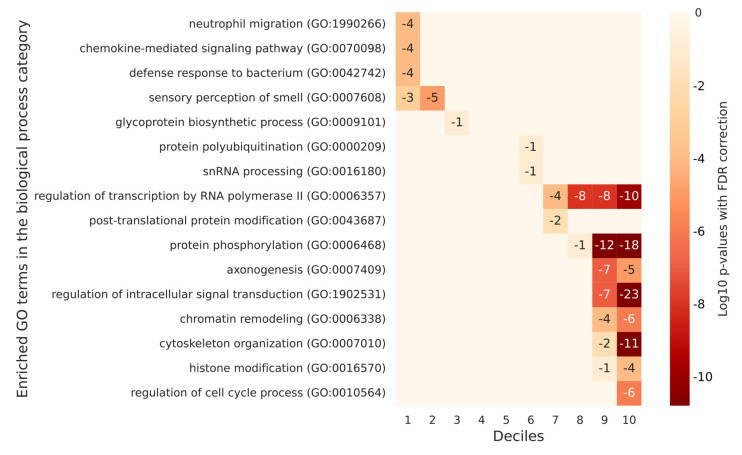
A heatmap of selected GO enrichments identified for human proteins grouped by decile based on their number of phosphorylation sites. The figure lists the significant GO enrichments by biological process terms for each decile of human proteins defined according to their degree of phosphorylation. Each table row represents a different GO term, and each column represents a different decile. The numbers in the cells are the log10 *p*-values with FDR correction of the enrichment of a decile with a specific GO term. Empty cells are associated with a log10 *p*-value of 0, which corresponds to a *p*-value of 1 and is not significant. In humans, non-phosphorylated proteins (those in the first two deciles) have a role in the immune response and perception. The most phosphorylated proteins (belonging to the last four deciles) have multiple regulatory roles and functions related to development and morphogenesis.

**Figure 7 ijms-23-14429-f007:**
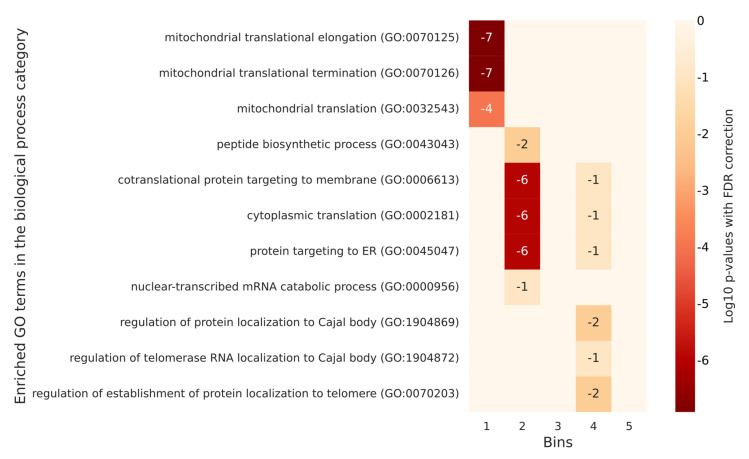
A heatmap of selected GO enrichments identified for *D. melanogaster* proteins grouped according to their number of phosphorylation sites. The figure lists the significant GO enrichments by biological process terms for each bin of *D. melanogaster* proteins defined according to their degree of phosphorylation. Each table row represents a different GO term, and each column represents a different bin. The numbers in the cells are the log10 *p*-values with FDR correction of the enrichment of a bin with a specific GO term. Empty cells are associated with a log10 *p*-value of 0, which corresponds to a *p*-value of 1. In the flies, the less phosphorylated proteins (those in the first two bins) have a role in protein formation and transport. Few annotations are associated with the most phosphorylated proteins.

**Figure 8 ijms-23-14429-f008:**
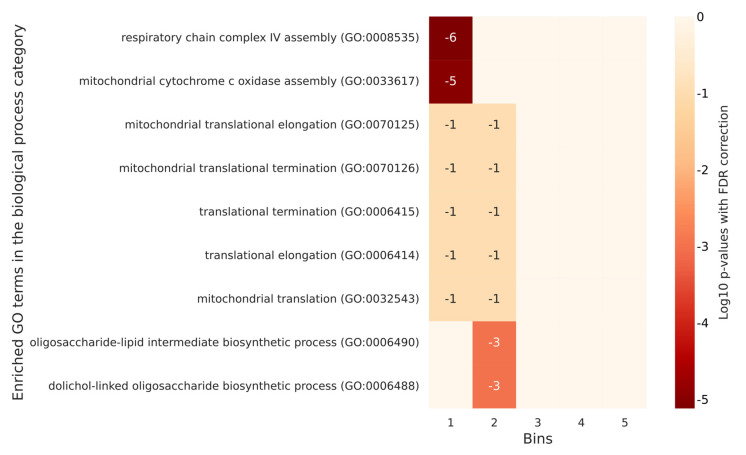
A heatmap of selected GO enrichments identified for *S. cerevisiae* proteins grouped by quintile based on their number of phosphorylation sites. The figure lists the significant GO enrichments by biological process terms for each quintile of *S. cerevisiae* proteins defined according to their degree of phosphorylation. Each table row represents a different GO term, and each column represents a different quintile. The numbers in the cells are the log10 *p*-values with FDR correction of the enrichment of a bin with a specific GO term. Empty cells are associated with a log10 *p*-value of 0, which corresponds to a *p*-value of 1. In the yeast, the less phosphorylated proteins (those in the first two quintiles) have a role in protein formation and transport. Oligosaccharide biosynthetic processes are associated with proteins in the second quintile, and no significant annotation are found for the most phosphorylated proteins.

**Figure 9 ijms-23-14429-f009:**
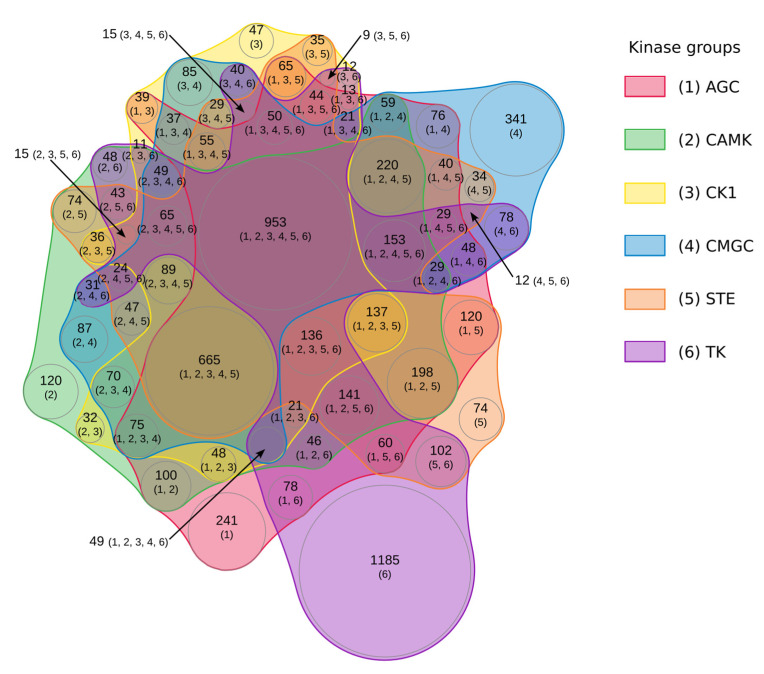
An euler diagram of the proteins phosphorylated by six major kinase groups. Euler diagram representation of the overlapping sets of proteins phosphorylated by the groups of kinases AGC, CAMK, CK1, CMGC, STE, and TK. In the figure, each kinase group is identified by a number between brackets (from 1 for AGC to 6 for TK). The number of proteins corresponding to each region of the diagram is specified, followed by the identifiers of the kinase groups that phosphorylate them. In dense areas, the content of some regions is taken out of the body of the diagram and linked to the corresponding region by arrows. The area of each region in the diagram is proportional to the number of elements included.

**Figure 10 ijms-23-14429-f010:**
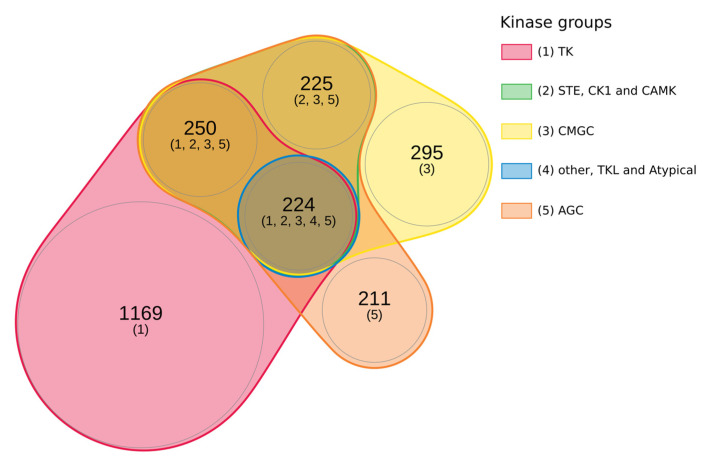
A filtered Euler diagram of the significant groups of proteins phosphorylated by five major kinase groups. The Euler diagram represents the overlapping sets of proteins phosphorylated by the most prominent kinase groups, i.e., those that target more than 2% of the total proteins. The STE, CK1, and CAMK groups, which phosphorylate exactly the same proteins, are grouped in a same category. The same goes for the groups of kinases Other, TKL, and Atypical. In the figure, each kinase group is identified by a number between brackets (from 1 for TK to 5 for AGC). The number of proteins corresponding to each region of the diagram is specified, followed by the identifiers of the kinase groups that phosphorylate them. The area of each region in the diagram is proportional to the number of elements include.

**Figure 11 ijms-23-14429-f011:**
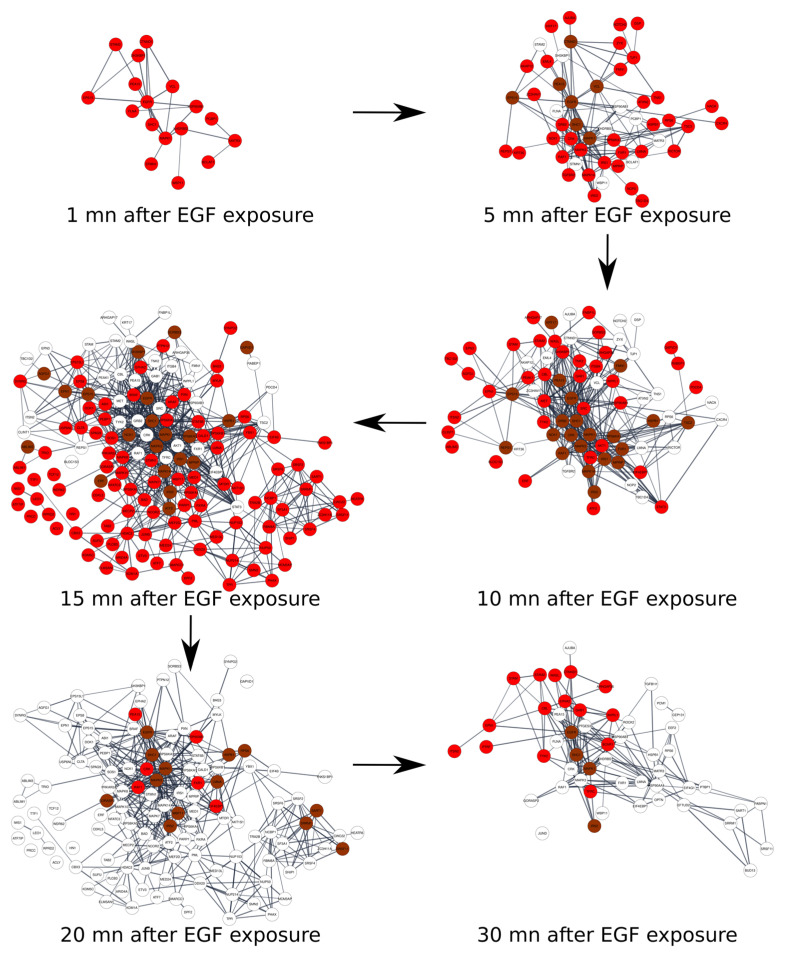
An interaction graph between the proteins included in the most striking module identified by AMINE for each of the time points after EGF exposure. The proteins in red and those in brown are those that were differentially phosphorylated compared to the control. The proteins in red represent those that were differentially phosphorylated at this time point compared to the previous time point (newly phosphorylated proteins). The proteins in brown are those that were differentially phosphorylated but were already differentially phosphorylated at the previous time point. Finally, the proteins with a white background are those that were no longer differentially phosphorylated at this time point but were at the previous time point (or back to normal).

**Table 1 ijms-23-14429-t001:** Statistics about the proteins of the seven studied species.

	*H. sapiens*	*M. musculus*	*D. rerio*	*D. melanogaster*	*C. elegans*	*S. cerevisiae*	*A. thaliana*
**All proteins**							
count (%)	157,160 (100%)	169,976 (100.00%)	175,752 (100.00%)	95,752 (100.00%)	81,312 (100.00%)	40,952 (100.00%)	122,984 (100.00%)
avg. length	567.70	545.88	569.25	553.17	494.47	519.80	453.27
avg. weight (dalton)	63,239.35	60,899.63	63,738.36	61,708.17	55,752.23	58,769.06	50,616.29
avg. nb of phos. sites	26.33	12.16	0.40	3.00	1.78	15.60	3.10
avg. nb of phos. sites/1000aa	44.08	20.14	0.68	4.44	2.87	28.76	6.79
**Non-phosphorylated**							
count (%)	8760 (5.57%)	34,880 (20.52%)	158,816 (90.36%)	58,704 (61.31%)	58,504 (71.95%)	5312 (12.97%)	56,992 (46.34%)
avg. length	189.81	286.04	535.28	408.51	403.88	275.86	349.89
avg. weight (dalton)	21,073.58	32,059.07	60,001.19	45,756.87	45,696.83	31,230.69	39,157.83
**Non-phos. & homologous**							
count (%)	4752 (3.02%)	24,856 (14.62%)	84,888 (48.30%)	14,392 (15.03%)	22,768 (28.00%)	3672 (8.97%)	17,920 (14.57%)
avg. length	217.63	308.83	537.63	467.38	462.18	306.98	394.12
avg. weight (dalton)	24,179.38	34,604.76	60,277.16	52,470.09	52,047.64	34,790.76	44,152.47
**Non-phos. & non-homologous**							
count (%)	4008 (2.55%)	10,024 (5.90%)	73,928 (42.06%)	44,312 (46.28%)	35,736 (43.95%)	1640 (4.00%)	39,072 (31.77%)
avg. length	156.82	229.55	532.57	389.38	366.73	206.17	329.60
avg. weight (alton)	17,391.26	25,746.64	59,684.32	43,576.50	41,650.63	23,259.61	36,867.08
**Phosphorylated**							
count (%)	148,400 (94.43%)	135,096 (79.48%)	16,936 (9.64%)	37,048 (38.69%)	22,808 (28.05%)	35,640 (87.03%)	65,992 (53.66%)
avg. length	590.01	612.97	887.84	782.39	726.85	556.15	542.56
avg. weight (alton)	65,728.38	68,345.88	98,783.34	86,983.62	81,544.98	62,873.55	60,512.04
avg. nb of phos. Sites	27.88	15.31	4.14	7.75	6.34	17.93	5.78
avg. nb of phos. Sites/1000aa	46.69	25.34	7.02	11.49	10.25	33.05	12.65
**Phos. & homologous**							
count (%)	135,000 (85.90%)	118,872 (69.93%)	10,576 (6.02%)	16,248 (16.97%)	11,712 (14.40%)	28,584 (69.80%)	24,152 (19.64%)
avg. length	603.56	623.26	824.95	814.92	702.19	561.02	554.76
avg. weight (alton)	67,246.82	69,484.93	91,920.53	90,881.20	78,902.92	63,482.53	61,901.72
avg. nb of phos. Sites	28.73	15.74	4.05	8.11	5.57	15.97	5.65
avg. nb of phos. Sites/1000aa	46.93	25.43	7.18	11.92	9.68	30.14	11.75
**Phos. & non-homologous**							
count (%)	13,400 (8.53%)	16,224 (9.54%)	6360 (3.62%)	20,800 (21.72%)	11,096 (13.65%)	7056 (17.23%)	41,840 (34.02%)
avg. length	453.47	537.58	992.41	756.98	752.89	536.44	535.52
avg. weight (alton)	50,430.62	60,000.12	110,195.46	83,939.00	84,333.71	60,406.56	59,709.85
avg. nb of phos. Sites	19.35	12.13	4.30	7.47	7.14	25.86	5.85
avg. nb of phos. Sites/1000aa	44.19	24.63	6.76	11.15	10.84	44.87	13.18

The table shows the average length and weight of proteins distributed according to different criteria (including the presence of a phosphorylation site and the existence of homologous proteins in another species). For phosphorylated proteins, the average number of phosphorylation sites and the average number of phosphorylation sites per 1000 amino acids are also specified.

**Table 2 ijms-23-14429-t002:** Interaction data for human proteins depending on their rate of phosphorylation.

Decile	1	2	3	4	5	6	7	8	9	10
ranges	[0, 2]	[3, 5]	[6, 8]	[9, 11]	[12, 14]	[15, 19]	[20, 26]	[27, 37]	[38, 61]	[62, 2518]
nb nodes	2368	2130	1985	1818	1538	2042	1964	1892	1984	1924
nb edges	1455	1607	2021	2217	1499	2840	3819	4240	7688	9188
avg. neighbors	1.229	1.509	2.036	2.439	1.950	2.781	3.889	4.482	7.750	9.551
nb connected	120	135	89	83	80	76	52	39	33	22
% singletons	67.95	55.77	48.51	46.97	48.57	42.36	37.58	33.51	23.14	15.96

Human proteins are divided in deciles depending on the number of their phosphorylation site. Interaction data were collected using the String database with a confidence score > 0.7. For each decile, the number of proteins (nb nodes), the number of interactions between the proteins (nb edges), the average number of neighbors of each protein with interactions (avg. neighbors), the number of connected components (nb connected), and the percentage of proteins with no interactions (% singletons) are given.

**Table 3 ijms-23-14429-t003:** Interaction data for *D. melanogaster* proteins depending on their rate of phosphorylation.

Ranges	0	1	[2, 3]	[4, 8]	[9, 454]
nb nodes	7338	1111	1273	1110	1137
nb edges	28,323	2232	2965	3156	4514
avg. neighbors	7.72	4.02	4.66	5.69	7.94
nb connected	123	38	36	15	17
% singletons	35.12	41.49	34.72	29.46	19.88

For this species, the non-phosphorylated proteins constitute the first bin, and the remaining proteins are divided into quartiles. Interaction data were collected using the String database with a confidence score > 0.7. For each bin, the number of proteins (nb nodes), the number of interactions between the proteins (nb edges), the average number of neighbors of each protein with interactions (avg. neighbors), the number of connected components (nb connected), and the percentage of proteins with no interactions (% singletons) are given.

**Table 4 ijms-23-14429-t004:** Interaction data for *S. cerevisiae* proteins depending on their rate of phosphorylation.

Ranges	[0, 1]	[2, 5]	[6, 11]	[12, 23]	[24, 227]
nb nodes	1020	1029	1072	987	1011
nb edges	3190	3628	7945	8366	5508
avg. neighbors	6.25	7.05	14.82	16.95	10.90
nb connected	29	29	17	16	8
% singletons	14.12	11.47	20.15	6.18	12.56

For this species, proteins are divided in quintiles depending on the number of their phosphorylation site. Interaction data were collected using the String database with a confidence score > 0.7. For each quintile, the number of proteins (nb nodes), the number of interactions between the proteins (nb edges), the average number of neighbors of each protein with interactions (avg. neighbors), the number of connected components (nb connected), and the percentage of proteins with no interactions (% singletons) are given.

## Data Availability

Data is contained within the article or Appendix A.

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
