# Peer review of "Evolutionary Divergence of Phosphorylation to Regulate Interactive Protein Networks in Lower and Higher Species"

_ijms, 2022, doi:10.3390/ijms232214429_

Round 1
Reviewer 1 Report
In this paper, Pasquier and Robichon have highlighted the importance of phosphorylation in signaling across different species by performing an extensive analysis of identifying the commonalities and differences of proteins undergoing phosphorylation. Targeting a wide number of proteins and its orthologues, phosphorylation plays a key role in shaping of different organisms' cell growth, function and development. This is reflected in the different analyses presented in this paper.
1. The authors have discussed at length on phosphorylation and its effects, however, the need for this study remains unclear. There are already many evolutionary and comparative studies on phosphoproteomes. The hypothesis/aim should be written clearly describing what has not been done, and how they resolve it by this study.
2. Although the findings of this paper can be of interest to multiple research groups, its significance is impeded by the constrictions of the numbers and global comparisons, and importantly in the lack of a significant take-home message. The discussion section needs to be improved to highlight the importance of the findings of this paper, and how other researchers can benefit from this. A general picture of phosphorylation has been painted, but some specifics of what these comparisons entail, and what can be expected from other phosphoproteome studies can be commented.
3. Authors have acknowledged that they have not focused on the sequences, but the amino acid sequence or the fold/motifs play a central role in successful phosphorylation events. While authors have extensively compared homologs globally, some pathways remain conserved across prokaryotes and eukaryotes, and highlighting some of these may enhance the significance of the paper to wider scientific community.
4. The phosphoproteome and list of proteins were procured via different databases. These often list sites of stable phosphorylation, while large macromolecular complexes regulate signaling events through transient phosphorylation. Can the authors comment on this? A short paragraph in the discussion, showing some of the implications of transient phosphorylation related to this study enhances the efforts from this paper.
5. On page 7, line 226-228, proteins with less than 500kDa with less than 500 phosphorylation sites were selected. Please specify what the number 500 represents. Is it an average or a fraction or a protein with ~500 phosphosites? This clarification across other comparisons is also needed.
6. At multiple locations, the words are misspelled and require correction. Some sentences in the introduction and discussion are very long and hard to understand. A shorter sentence structure, with appropriate punctuation marks, will allow readers a better understanding.
Reviewer 2 Report
The manuscript entitled: "Prominence of phosphorylation to orchestrate ..." is quite an interesting work, which can be published with slight modifications. Most of the errors are practically visual corrections. In terms of content, the work does not contain too many errors.
Below are my comments:
1. Please re-edit the title slightly. Sounds a bit weird.
2. The first two sentences of the abstract must be rewritten. They're a little distracting.
3. Section: keywords should probably be supplemented - the work is probably much more extensive.
4. "... PhosphoSitePlus 106 (https://www.phosphosite.org), Phospho.ELM .." and others - can't these citations be normal citations? I don't know why entered website addresses.
5. It seems to me that the literature should be supplemented a bit, as for such an extensive work there are probably not enough citations. There is much more information on the Internet, even after typing these keywords as provided by the authors.
6. Please rewrite the last paragraph from the introduction. And to be more precise about the purpose of the work. Because so far there is a lot, but what is it?
7. Fig 1 - I think it could be nicer.
8. Fig 3 - probably could be nicer too.
9. Fig 5 - probably could be nicer too.
10. Fig. 6 - also careless. Dropped captions on the y-axis, please correct it.
11. Figs. 7 and 8 - also careless. Dropped captions on the y-axis, please correct it.
12. I miss a summary section. It is unacceptable for such extensive work. Please correct it, or rewrite it at the end of the discussion and fill it in slightly.
Overall a nice job. A few unnecessary bugs spoil the image a bit, but they are easy to fix.
Round 2
Reviewer 2 Report
The authors improved the manuscript very well. I recommend accepting the manuscript.